# Beauty-in-averageness and its contextual modulations: A Bayesian statistical account

**Chaitanya K. Ryali**
Department of Computer Science and Engineering
University of California San Diego
9500 Gilman Drive La Jolla, CA 92093
`rckrishn@eng.ucsd.edu`

**Angela J. Yu**
Department of Cognitive Science
University of California San Diego
9500 Gilman Drive La Jolla, CA 92093
`ajyu@ucsd.edu`

## Abstract

Understanding how humans perceive the likability of high-dimensional "objects" such as faces is an important problem in both cognitive science and AI/ML. Existing models generally assume these preferences to be fixed. However, psychologists have found human assessment of facial attractiveness to be context-dependent. Specifically, the classical Beauty-in-Averageness (BiA) effect, whereby a blended face is judged to be more attractive than the originals, is significantly diminished or reversed when the original faces are recognizable, or when the blend is mixed-race/mixed-gender and the attractiveness judgment is preceded by a race/gender categorization, respectively. This "Ugliness-in-Averageness" (UiA) effect has previously been explained via a qualitative disfluency account, which posits that the negative affect associated with the difficult race or gender categorization is inadvertently interpreted by the brain as a dislike for the face itself. In contrast, we hypothesize that human preference for an object is increased when it incurs lower encoding cost, in particular when its perceived *statistical typicality* is high, in consonance with Barlow's seminal "efficient coding hypothesis." This statistical coding cost account explains both BiA, where facial blends generally have higher likelihood than "parent faces", and UiA, when the preceding context or task restricts face representation to a task-relevant subset of features, thus redefining statistical typicality and encoding cost within that subspace. We use simulations to show that our model provides a parsimonious, statistically grounded, and quantitative account of both BiA and UiA. We validate our model using experimental data from a gender categorization task. We also propose a novel experiment, based on model predictions, that will be able to arbitrate between the disfluency account and our statistical coding cost account of attractiveness.

## 1  Introduction

Humans readily express liking and disliking for complex, high-dimensional "objects", be they faces, movies, houses, technology, books, or life partners, even if they cannot verbalize exactly why. Understanding how these preferences arise is important for both cognitive science, and for AI systems that interact with humans. In particular, face processing presents a prime case study for complex information processing in humans. Human, including very young babies [1], efficiently perform sophisticated computational tasks based on a brief glimpse of a face, such as recognizing individuals, identifying emotional states, and assessing social traits such as attractiveness [2]. The last phenomenon has an obvious impact on real-life decisions such as dating, employment, education, law enforcement, and criminal justice [3]. Existing models of human preferences, in both machine learning and cognitive science, have generally assumed social processing of faces (e.g. attractiveness judgment) to be a *fixed* function of the underlying face features [4, 5, 6, 7, 8, 9]. However, a

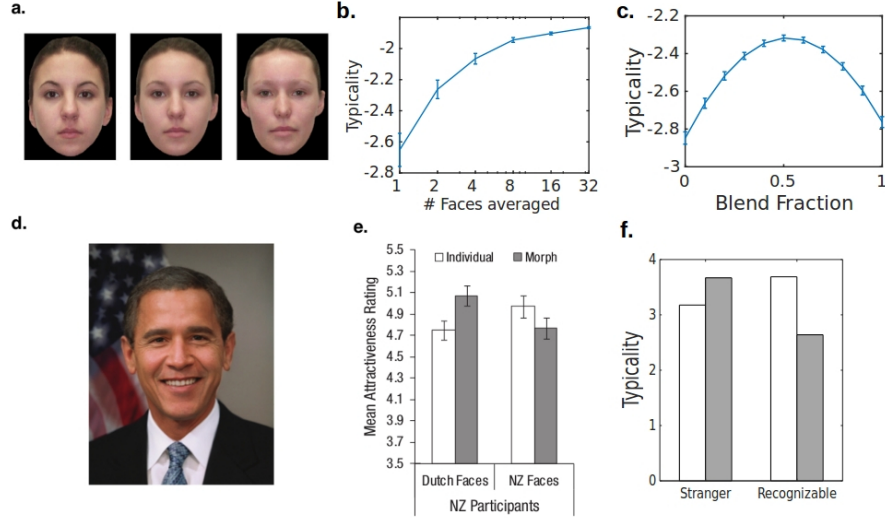

Figure 1: BiA, UiA in celebrity morphs. (a) The middle face, a 50% blend of the faces on the left and right, is generally judged by human subjects to be more attractive than either "parent" face (from [12]). (b) Simulated typicality increases with increasing number of faces used in the blend. (c) Simulated typicality of facial blends increases as the "parent" faces are more evenly represented in the blend. (d) Example image (from [11]) depicting a morph of two recognizable faces (here, Bush and Obama). (e) Blends of recognizable individuals are rated by human subjects as less attractive than individual recognizable faces, while blends of stranger faces are rated as more attractive (adapted from [11]). (f) Simulated typicality has similar pattern as data (e). A constant offset of 6 was added to produce positive values. Simulation parameters: $d = 60$, $d_{\text{race}} = 1$, $s = 2$, $\sigma_0 = 1$, $\sigma_r = 0.5$ and $\mu = 1$, $|K| = 50$, $\sigma_{\text{sal}} = 0.2$, $\sum_{k=1}^{|K|} p_k = 0.05$, all simulations in 2-d subspace, corresponding to a random subspace or a distinctive feature subspace.

series of recent psychology experiments have indicated that facial preferences in the brain are not fixed, but rather systematically dependent on what other face-processing task the observer is also performing. Specifically, these experiments show that a classical phenomenon known as the beauty-in-averageness (BiA) effect (see Figure 1a), whereby blends of multiple faces are usually found to be more attractive than the originals [10], can be suppressed or even reversed (termed Ugliness-in-Averageness or UiA), when the facial blends are created from recognizable faces [11] (see Figure 1d;e), or when attractiveness judgment of a mixed-race/mixed-gender blend is preceded by a racial/gender categorization task [12, 13] (see Figure 2a;b), respectively.

The facial BiA effect has long been seen as an example of human preference for highly prototypical stimuli over more unusual stimuli [14]. Early accounts explained this phenomenon as reflecting a biological predisposition to interpret prototypicality as a cue to mate value or reproductive health [15, 16]. However, this mate-value account cannot explain human preference for prototypicality in a variety of natural and artificial objective categories such as dogs, birds, fish, automobiles, watches, and even synthetic dot patterns [17, 18, 14]. Moreover, it cannot explain why attractiveness of facial blends should depend sensitively on the behavioral context, e.g. when required to do racial or gender discrimination [12, 13]. To explain this diverse array of phenomena, a more parsimonious account based on processing fluency has been proposed: prototypes are processed more "fluently" and humans prefer more fluently processed stimuli [14, 12].

While conceptually appealing, this fluency account does not explain what "fluency" really means computationally, nor in what sense it may be advantageous for the organism. We address issues, by hypothesizing that human preference for an object is increased when it incurs lower *coding cost*, in particular when its perceived *statistical typicality* is high. This hypothesis is consonant with ideas from information theory [19] and Barlow's "efficient coding hypothesis," which stipulates that neural encoding should be organized so as to minimize the energy expenditure needed to represent the sensory environment, in particular using fewer spikes (expending less energy) to encode more probable stimuli [20, 21, 22, 23]. Efficient coding is necessary given that the brain accounts for

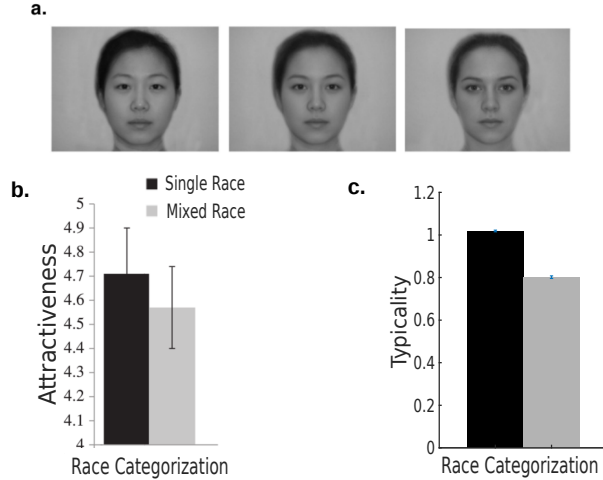

Figure 2: UiA of biracial blends. (a) Example stimuli used in [12], with the middle face being a 50% blend of the Asian and Caucasian faces on either side. (b) Mean attractiveness ratings for single-race (left), and mixed-race blends (right), when race categorization preceded the attractiveness rating (from [12]). (c) Simulated typicality exhibits similar pattern as human data. A constant offset of 2 was added to produce positive values. Simulation parameters same as in Figure 1.

20% of the adult human body's energy expenditure, but only 2% of its weight [24]. In the context of faces and other complex objects, we suggest that the brain represents each new stimulus by first finding the closest previously learned representative (prototype) [25], and then use neural activations to encode the discrepancy between the features of the inferred prototype and the current stimulus, perhaps via predictive coding [26], whereby the top-down inputs instantiate prototypical expectations and bottom-up activation encode additional prediction errors. Without delving into a detailed neural implementation of such a computational process [27], we broadly quantify this "coding cost" in terms of category-conditional log likelihood (Section 2): the smaller the likelihood of the stimulus conditioned on the inferred category, the more encoding cost it incurs. To summarize, we recast "processing fluency" into this statistically grounded notion of "category-conditional likelihood", and propose that a statistically likely stimulus is preferable for humans due to lower neural coding cost, or less energy required to encode the prediction error.

Besides coding efficiency, we make a second major assumption, which is that the brain uses attentional mechanisms to focus or project its face representation into a subset (subspace) of task-relevant features, and that statistical typicality and thus coding cost are redefined in this projected representation. This is consistent with a large body of work showing that perceptual tasks are often performed in a low-dimensional, task-relevant subspace [28] focusing on informative dimensions in high dimensional data, using attentional modulation that can be either top-down goal-directed [29] or bottom-up saliency-based [30]. It is also consistent with a body of work showing that people often use category membership to predict features and reason about members of the category [31, 32].

We will give a short intuitive explanation of how our theory explains both BiA and UiA. In BiA, assuming the population distribution of faces is unimodal (something we will verify on real face image data in Section 3), facial blends will generally have higher likelihood than "parent faces", and therefore higher attractiveness ratings. On the other hand, when the behavioral context or task induces the brain to restrict face representation to a task-relevant subset of features, such as the subspace that maximally separates male and female faces in a gender discrimination task, then statistical typicality is *redefined* within that subspace to be particularly unfavorable toward stimuli that are situated in between categories in the task-relevant subspace but close to the center in the original undifferentiated representation. This scenario explains why bi-racial or bi-gender blends are generally perceived as more attractive than their "parent faces", but that effect is diminished or even reversed when there is an explicit race- or gender-discrimination task [12, 13], respectively. Similarly, our model predicts that blends of familiar faces should be viewed as less attractive [11]: we hypothesize familiar faces are represented by their statistically distinctive features (features that differentiate them from generic

faces), and the recognition of familiar faces leads the attentional system to focus on the subset of distinctive features, within which the facial blends are statistically relatively less likely.

The rest of the paper is organized as follows. In Section 2, we formally define coding cost, statistical typicality, and attentional focusing in face processing. In Section 3.1, we use an abstract statistical model to illustrate our theory, and show how BiA and context-induced UiA can arise under our statistical assumptions. In Section 3, we use a real face image data set, and an explicit face representation commonly used in machine vision, to fit a parametric distributions of faces. We then verify statistical assumptions of our abstract model using this face dataset, as well as making detailed predictions of attractiveness (as a percentages of bi-racial and bi-gender blend) based on our measure of statistical typicality, the category-conditional log likelihood. Then, using actual face stimuli used in a gender-discrimination task [12] and projecting them into our face representation, we show that our predicted facial attractiveness of these stimuli significantly correlate with subjects' actual attractiveness ratings. Next, we propose an experiment to disambiguate the disfluency and statistical typicality accounts of UiA (Section 4). Finally, we conclude in Section 5 with a discussion on the limitations of our model and future directions of research.

## 2 A Formal Representation of Faces and Attractiveness

We assume humans have an internal $d$-dimensional representation of faces $\mathcal{X}$ [33, 5, 34], in which each face is represented by a vector $\mathbf{x} = (x_1, \ldots, x_d)$ of $d$ real-valued features. We also assume that this face space is endowed with a probability distribution $p_\mathcal{X}(\mathbf{x})$ representing the perceived distribution of faces in the environment [35], which in general can be a complex mixture distribution with different components corresponding to different subgroups of the population (e.g. different races, genders, other subtypes). In general, we assume facial attractiveness is proportional to log likelihood of the face, $\log p_\mathcal{X}(\mathbf{x})$. In the absence of any categorization task, explicit (e.g. race, gender) or implicit (e.g. individual recognition), we assume more likely faces are preferred. This explains the BiA effect, as long as $p_\mathcal{X}(\mathbf{x})$ is approximately a single-peaked distribution (e.g. Gaussian), since the average of two points drawn from such a distribution will probably have a higher likelihood than the original two points. We will show in Section 3 that the empirical distribution of a large sample of real face images is indeed approximately Gaussian.

It may seem puzzling that $p_\mathcal{X}(\mathbf{x})$ is both approximately Gaussian and a mixture distribution. The reason is that the different components differ from each other only on a small subset of features. For example, male and female faces may have indistinguishable distributions along most featural dimensions, but be quite distinct in features that are especially gender discriminating. We will see that real face data indeed exhibit this property in Section 3.

When the observer performs a categorization task, such as race discrimination, we assume the attractiveness of a face stimulus is proportional to the category-conditional log-likelihood $\log p_\mathcal{X}(\mathbf{x}|c)$, where $c$ is the estimated category, among those of potential interest, based on the general distribution $p_\mathcal{X}(\mathbf{x})$. For example, $p_\mathcal{X}(\mathbf{x})$ can be viewed as approximately a mixture of two Gaussians (male and female) in the gender discrimination task (see Section 3). We propose that the brain first uses Bayesian estimation to estimate the gender of a face (e.g. $c = $ male), and then the attractiveness of the face would be proportional to $p_\mathcal{X}(\mathbf{x}|c = \text{male})$, which is inversely related to the coding cost necessary to represent $\mathbf{x}$ on top of knowing its category.

An additional wrinkle is that we assume, in the context of a particular task, the brain projects the face space and its distribution into a task-relevant subspace $\tilde{\mathcal{X}} \subseteq \mathcal{X}$. Denoting the projection of a face $\mathbf{x}$ into a subspace $\tilde{\mathcal{X}}$ as $\tilde{\mathbf{x}}$, we redefine statistical typicality in the subspace $\tilde{\mathcal{X}}$ as $\log p_{\tilde{\mathcal{X}}}(\tilde{\mathbf{x}}|c)$, the log-likelihood of $\tilde{\mathbf{x}}$ constrained to the subspace $\tilde{\mathcal{X}}$, conditioned on the Bayesian estimated category $c$. In a race- or gender-categorization task, we assume the brain projects the face space into a race- or gender-informative subspace $\tilde{\mathcal{X}}_{\text{cat}}$, respectively. Bi-racial/bi-gender blends are statistically atypical of both categories in this projected space, where the different race or gender categories are clearly distinct, thus resulting in low attractiveness ratings.

We also use this projection idea to model BiA in blends of familiar or recognizable faces. We assume that the brain by default performs recognition, a categorization task, whereby recognizable faces have their own modes, plus there is a general distribution for all unfamiliar faces. It has been suggested that people encode familiar faces using features that are most distinctive/salient, as this is

not only computationally efficient but may also boost recognition [36]. Accordingly, we assume that a familiar face is represented by its $s$ statistically most distinctive (atypical) features: we assume $\mathbf{x}_f$ is represented by its veridical value, if it is among the top $s$ $z$-scored dimensions, and $0$ otherwise. We assume that a blend of two recognizable faces $\mathbf{x}$ induces an implicit categorization in the subspace $\tilde{\mathcal{X}}_{\text{sal}}$ spanned by a subset of the distinctive features of the parent faces (an alternative is to project to the subspace spanned by the blend's own distinctive features, an approach that yields similar results in our simulations, which are not shown here). Similar to explicit categorization tasks, $c$ is the a posteriori (i.e., after classification) most probable identity, and statistical typicality in this case is $\log p_{\tilde{\mathcal{X}}_{\text{sal}}}(\tilde{\mathbf{x}}_{\text{sal}}|c)$. The attractiveness rating is low for the blend $\mathbf{x}$ in this subspace, because it is sufficiently unlike either of the parent faces (low conditional likelihood), but also low likelihood relative to the general distribution given that this is by definition the subspace of facial features that are distinctive (statistically unlikely). Relatedly, people might not compute the statistical typicality of a face with respect all the underlying features in the absence of an implicit or explicit categorization task, and may do some only for a random subset of features.

## 3 Demonstrations

We will first present a simple abstract model in section 3.1 that captures both BiA and as well as UiA in various contexts. The simplicity of this model is deliberate, in that it is meant to be both expository, as well as demonstrating the generality of our proposal, since BiA and UiA are not specific to faces but emerge for other natural and synthetic objects [37, 14, 38]. In section 3.2, we use a data-based face space representation for further validation.

### 3.1 Abstract Model

#### 3.1.1 Generative Model

We assume that humans internally represent each face $\mathbf{x} = (x_1, \ldots, x_d) \in \mathbb{R}^d$ as generated from a mixture of Gaussians, whereby the components can either correspond to well-known faces $\{f_i\}$ (assume $K$ of these) or demographic subgroups $\{h_r\}$ (assume $G$ of these, e.g. gender, race),

$$\boldsymbol{X} \sim \sum_{k=1}^{|K|} p_k f_k(\boldsymbol{x}) + (1 - \sum_{k=1}^{|K|} p_k) g(\boldsymbol{x}), \tag{1}$$

$$g(\boldsymbol{x}) = \sum_{r=1}^{|G|} q_r h_r(\boldsymbol{x}), \tag{2}$$

where $h_r(\boldsymbol{x}) = \mathcal{N}(\boldsymbol{x}; \boldsymbol{\mu}_r, \boldsymbol{\Sigma}_r)$, $f_k(\boldsymbol{x}) = \mathcal{N}(\boldsymbol{x}; \boldsymbol{\mu}_k, \boldsymbol{\Sigma}_k)$ and $\sum_{k=1}^{|K|} p_k << 1$ as the number of known faces should be much fewer than unknown faces. We assume that the distributions of the mixture components $h_r$ differ only in a small number of dimensions, $1, \ldots, d_{\text{race}}$ and are identical on the other $d_{\text{other}} := d - d_{\text{race}}$ dimensions. Specifically, we assume $\boldsymbol{\mu}_{r,d_{\text{race}}+1:d} = \mathbf{0} \in \mathbb{R}^{d_{\text{other}}}$ and

$$\boldsymbol{\Sigma_r} = \begin{bmatrix} \sigma_r^2 \mathbb{1}_{d_{\text{race}} \times d_{\text{race}}} & \mathbf{0} \\ \mathbf{0} & \sigma_0^2 \mathbb{1}_{d_{\text{other}} \times d_{\text{other}}} \end{bmatrix}, \tag{3}$$

where $\mathbb{1}_{n \times n}$ is an identity matrix of dimensions $n \times n$. For simplicity, we assume $|G| = 2$ and set $\boldsymbol{\mu_1} = -\boldsymbol{\mu_2} = \boldsymbol{\mu}$, where $\boldsymbol{\mu}_{1:d_{\text{race}}} = [\mu, \ldots, \mu] \in \mathbb{R}^{d_{\text{race}}}$. We also set the prior/mixture probability distribution $\boldsymbol{q}$ to be uniform.

**Approximation**. Note that since the statistics of $h_r$ differ only in a small number of dimensions $d_{\text{race}} << d$, the mixture $g(\boldsymbol{x}) = \sum_{r=1}^{|G|} q_r h_r(\boldsymbol{x})$ is well approximated by $\tilde{g}(\boldsymbol{x}) = \mathcal{N}(\boldsymbol{x}; \boldsymbol{\mu}_0, \boldsymbol{\Sigma}_0)$, where $\boldsymbol{\mu}_0 = \mathbf{0} \in \mathbb{R}^d$ and $\boldsymbol{\Sigma}_0 = \sigma_0^2 \mathbb{1}_{n \times n}$ and can be assumed to be used to perform inference except when demographic features bear relevance, thus simplifying computations and representation.

**Salient feature representation**. The mixture components $\{f_k\}$ represent known/recognizable faces, where the variance in each component corresponds to natural variability in a face, such as variations in pose or expressions. For each face $k$, we assume subjects encode/represent only $s$ distinctive features (relative to the assumed generative distribution) as described in the previous section, denoted by $i_1^k, \ldots, i_s^k$ (the variance along these dimensions is denoted as $\sigma_{\text{sal}}^2$) and assume the same statistics

along other dimensions as $\tilde{g}(\boldsymbol{x})$, the approximate, assumed generative distribution for a generic, unfamiliar face.

**Recognition**. For simplicity, we assume the brain applies Bayes' rule to compute the posterior for each face **x**, and then picks the most probable category in each case via maximum a posteriori estimation.

### 3.1.2  Simulation Results

**BiA**. We first examine whether our abstract model can capture some nuances of the BiA effect. Our simulation shows that as the number of constituent faces that go into the blend increases, the typicality of the blend increases, so that the blend is expected to be perceived as increasingly more attractive (see Figure 1b). This is consistent with the finding that attractiveness of faces increases (decreases) when they are distorted towards (away from) the population mean [39, 4]. Additionally, it has been found that more evenly blended face images are perceived as more attractive [12], something that is also captured by the typicality measure in a simulation of the abstract model (Figure 1c).

**UiA: Familiar Faces**. In [11], participants from Netherlands and New-Zealand rated blends of local celebrities (people famous in one country but not the other). Blends of unknown celebrities were rated as more attractive than the "parent" face images (classic BiA), while blends of local celebrities were rated as less attractive relative to the constituent images: a reversal of BiA. An example image (from [11]) depicting a morph of two recognizable faces can be seen in Figure 1d, while Figure 1e shows BiA and its reversal in data from the study. As discussed in section 2, low statistical typicality of the blend in the distinctive feature subspace (here 1-d) results in UiA. Simulations qualitatively capture this effect in Figure 1f.

**UiA: Race Categorization**. In [12], participants rated mixed and single race blends on attractiveness after performing a race categorization task (Asian or Caucasian). An example of stimuli used in [12] is shown in Figure 2a. Data in Figure 2b shows that mixed-race morphs are rated as less attractive relative to single race morphs when a race categorization task preceded the attractiveness judgment. As previously hypothesized in section 2, low statistical typicality of a mixed race blend in the subspace of race informative features induces UiA. For simplicity, we assume this subspace is the one determined by Linear Discriminant Analysis (LDA). Simulations qualitatively capture the behaviour of attractiveness judgments in data (Figure 2c).

### 3.2  Data-Based Face Representation

We model faces using the Active Appearance Model (AAM), a well-established machine vision technique that reconstructs images well, generates realistic synthetic faces, produces a latent representation of only a few dozen features [40, 41, 42], and whose features seem to be encoded by face processing neurons in the monkey brain [43]. AAM assigns each face image a vector of *shape features*, which are just the $(x, y)$ coordinates of some consistently defined landmarks across all faces – in our case, we use the free software Face++ [1], which labels 83 landmarks (e.g. contour points of the mouth, nose, eyes). AAM also assigns each face image a vector of *texture features*, which are the grayscale pixel values of a warped version of the image after aligning the landmark locations to the average landmark locations across the data set (see Figure 3a for a schematic illustration of AAM). Consistent with standard practice for reducing the dimensional of AAM [40, 41, 42, 43], we perform principal component analysis (PCA) on each of the shape and texture features. In addition, to remove the correlation among shape and texture features, we then perform another PCA to get joint shape-texture features that are statistically uncorrelated with one another. We use the top 60 principal components (highest eigenvalues) ($d = 60$ for our face space). We train our version of AAM using a publicly available dataset of 597 face images [44], with neutral facial expression taken in the laboratory.

First, we *validate several assumptions* made in the abstract model. We find the distribution of faces learned from data is indeed approximately normal along a random face space (AAM) axis (Figure 3b), but a mixture of two Gaussians in race-informative (Figure 3c) or gender-informative subspaces (Figure 3d), found using LDA. We then use this face data-informed AAM representation to make nuanced *predictions* about facial attractiveness as a function of % blend between faces of different races or different genders [12, 13]. We first randomly drew 60 Asian and white face images (with

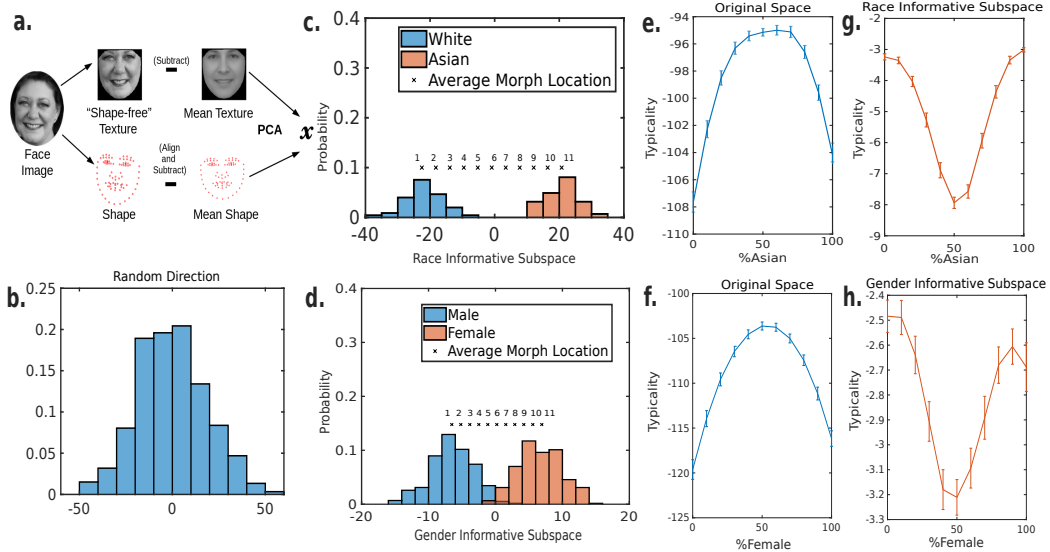

Figure 3: AAM-based face representation and model simulations. (a) AAM consists of shape and texture features; a joint PCA is then conducted over both types of features to remove correlations. (b) The empirical distribution projected in a random direction (1-d subspace) is normally distributed. (c) The empirical distribution of white and Asian faces projected into a race-informative subspace (1D subspace obtained by LDA) is approximately a mixture of two normal distributions. x: mean location of face images (60 total) for each % of blend, i.e. 1: 100% white and 0% Asian, ..., 11: 0% white and 100% Asian. (d) Analogous to (c) by projecting faces into a gender-informative subspace. x: mean location of face images (10 total) for each % of blend (male image: female image). (e) Model simulated typicality for actual face images (indexed by racial blend proportion) exhibits BiA as a function of racial blend when there is no categorization task. (f) Analogous to (e), but indexed by gender blend proportion. (g) Model simulated typicality for actual face images exhibits UiA when statistical typicality is measured in the race-informative subspace. (h) Analogous to (g) but measured in the gender-informative subspace.

replacement) from the face data set [44], and then blended them at 10% increments, from 100% of the white faces to 100% of the Asian faces, thus producing 11 morphs of each pair (see Figures 3c). Average predicted typicality, defined as category-conditional log likelihood, has an inverted U shape (BiA) relative to % racial blend in the original space (Figures 3e) or a random subspace (not shown), but a U shape (UiA) in the race-informative subspace (Figures 3g). Analogously, when 60 pairs of male and female faces are randomly drawn and then blended in different proportions, the model predicts typicality, and thus attractiveness, to have similar BiA (Figures 3f) and UiA effects (Figures 3h).

### 3.3 Experimental Validation: Attractiveness of Individual Faces

Using actual face stimuli used in the gender discrimination study [13], as well as subjects' actual attractiveness ratings, we can assess the ability of our model to predict the attractiveness of individual face images. In this study, subjects rated the attractiveness of blends from 10 unique pairs of male and female "parent" faces, in different proportions (10% increment, see Figure 4a), under either the control condition (no gender discrimination), or the experimental condition (following gender discrimination). We projected the stimuli into our AAM space (Figure 4b), and computed statistical typicality in the original/full space as well as the task informative subspace (Figure 4c;d). Even though this study used only 10 pairs of face images, we see that the model-predicted BiA/UiA effects are very similar to those based on a much larger random sampling from our face dataset (Figure 3f;h), and the predicted UiA pattern corresponds well to the reported attractiveness of the actual faces (Figure 4e). Moreover, we find that the difference in attractiveness ratings between experimental and control conditions correlate significantly with model-predicted typicality of individual face images ($r = 0.30$, $p = 0.0017$). We never expected the correlation to be close to 1, because there are clearly other determinants of facial attractiveness besides typicality, such as perceptual and conceptual

priming, contrast, clarity, and symmetry [14]. Indeed, female faces are generally found to be more attractive than male faces [14], which is why we plot the *difference* in attractiveness rating between experimental and control conditions – this removes any baseline effects of gender.

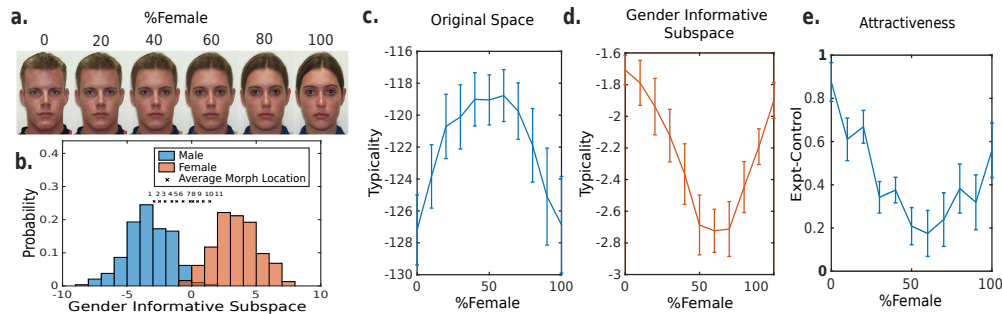

Figure 4: UiA of bi-gender blends induced by gender discrimination. (a) Example stimuli used in [13]: blends of varying proportions of male and female "parent" faces. (b) Analogous to Figure 3c;d, but using the actual experimental stimuli projected into our trained AAM. (c), (d) Analogous to Figure 3e;f;g;h, but using actual experimental stimuli projected into our trained AAM. (e) Difference in attractiveness ratings between experimental and control condition versus blend %.

## 4 Disentangling Disfluency and Typicality accounts

In the simulations and experiments considered so far, our statistical typicality account and the disfluency account make qualitatively similar predictions, because the difficulty of categorization and statistical typicality are monotonically related: the faces that are hardest to categorize are also the least likely given either category. To disambiguate these accounts, we need an experiment that dissociates categorization difficulty with statistical atypicality. We therefore suggest the following experiment that involves discriminating age, which is unimodally distributed (see the empirical distribution of age [45] in Figure 5a). The proposed experimental condition is to rate the attractiveness of faces after discriminating age: is the person older or younger than 37 years old? According to the statistical typicality account, attractiveness ratings would look like Figure 5b, having a shape similar to the population distribution (Figure 5a). In contrast, the disfluency account would qualitatively predict the difficulty of categorization to be the greatest and thus processing *fluency* and attractiveness to be lowest near the categorization boundary. Figure 5c illustrates this by simply joining two lines that decrease toward the categorization boundary. To summarize, our model predicts BiA in this experiment, while the disfluency account predicts UiA.

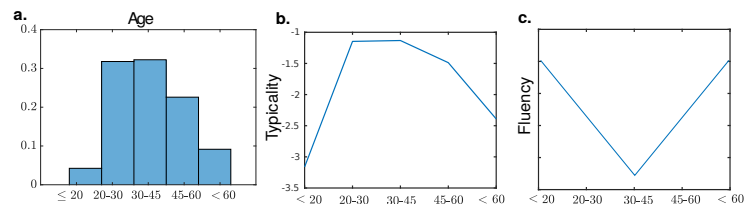

Figure 5: (a) Empirical distribution of ages in dataset [45]. (b) Predictions of our typicality based model. (c) Predictions based on a disfluency account.

## 5 Discussion

Most existing models of human preferences assume these preferences to be fixed and do not model contextual dependence. In this paper, we propose a statistically grounded model of human "liking", whereby the attractiveness of a stimulus depends on its neural coding cost, in particular how likely it is relative to its perceived category. This argument is based on information-theoretic considerations, in particular the coding cost associated with statistically unlikely stimuli, and is related to Barlow's "efficient coding hypothesis." Additionally, we assume that humans naturally project high-dimensional

data, such as faces, in a task-relevant manner to a low-dimensional subspace representation, either via top-down goal-directed specification (informativeness with respect to a particular discrimination problem, such as race or gender), or via bottom-up saliency (distinctiveness with respecive to the assumed generative distribution). Under our framework, therefore, the attractiveness of a stimulus is context-dependent for two different reasons: (1) the set of hypotheses under consideration in the Bayesian posterior computation is context-dependent (e.g. the two gender categories compose the hypothesis space in a gender discrimination task), and (2) the statistical distribution corresponding to the generative model changes according to the featural subspace that supports the current task (e.g. during a gender discrimination task, the face space and its distribution are defined only with respect to the subspace that is most-informative for gender discrimination).

While race and gender correspond to existing multi-modality in the distribution of faces, our theory would suggest that UiA can be produced in arbitrary cross-mode blends, if human participants can learn novel bimodal distributions of faces in an experimental setting – an experimentally testable prediction. Relatedly, it is worth making a distinction between statistical *typicality* as we define it, and *protoypicality*, as is usually conceived in the psychology literature [14]. Prototypicality implies there are clear modes in the stimulus distribution, and how *prototypical* a stimulus is presumably depends on how "close" it is to the closest mode. *Typicality*, as we define it, however, does not always depend monotonically on distance to the closest mode, and is well defined even for distributions that have no distinct modes (such as a uniform distribution). Separately, it is important to reiterate that we do not claim that statistical typicality or coding cost is the *only* determinant of attractiveness or "liking." Many other factors have been shown experimentally to be important for human preferences of complex objects, such as perceptual and conceptual priming, contrast, clarity, and symmetry [14].

In addition to providing a statistically grounded explanation of contextual dependence of human attractiveness judgment, our work also provides some general insight as to how high-dimensional data can be analyzed and stored *efficiently*: the system needs to be able to dynamically shift its subspace projection according to task demands, so as to reduce the need for representational and computational complexity at any given moment. Moreover, our work suggests two different ways to identify the appropriate subspace projection (and thus the appropriate form of complexity reduction). One is supervised, task-specified choice of hypothesis space, and thus the corresponding subspace projection that best discriminate the hypothesis – we argue this is what underlies context-induced decrease in the attractiveness of bi-racial/bi-gender faces during race- and gender-discrimination tasks, respectively. The other route is unsupervised, saliency-induced subspace projection, in which the statistically unlikely (relative to population distribution) features of a high-dimensional stimulus are privileged in their processing and encoding, and subsequent computations are performed within this subspace – this is what underlies our explanation of the UiA effect in celebrity-blend faces. The general idea of "tagging" high-dimensional data by their distinctive features seems like a good way to store and analyze complex data. Our work sheds light on one possible role played by attention: it is one way to dynamically construct subspaces that emphasizes feature dimensions that are most relevant or salient for performing the task at hand. There is a broad and confusing literature of attention in both psychology and neuroscience. A productive direction of future research would be to relate our hypothesized role of attention to that larger literature.

Though human attractiveness perception is interesting in itself, we are more fundamentally interested in a computational understanding of how the brain encodes and processes complex, high-dimensional data (e.g. faces), and how attention can dynamically alter the featural representation in a task-dependent manner. Faces are appealing because they are informationally rich, ecologically important, and for which we have a computationally tractable and neurally relevant [3] parametric representation (AAM). We therefore used faces to implement/test concrete ideas about information representation and its contextual modulation in this work. The attractiveness literature provides a convenient empirical test of our theory, but we expect that the dynamic representational framework we hypothesize here to also affect other cognitive processes, such as working memory, learning, decision-making, and problem-solving, in the sense that all these cognitive processes depend on what features are currently made salient by attentional mechanisms. For example, learning to memorize a set of items should be easier if one's attention is focused the features that make these items easiest to organize conceptually. Indeed, while the ecological benefits of "liking" based on encoding cost is rather generic and long-term, the benefits of cognitive expediency or accuracy derived from focusing on task-relevant features are acute and immediate. The latter points to a promising line of future research.

# 6    Acknowledgments

We thank Piotr Winkielman and Jamin Halberstadt for sharing the gender categorization data and helpful discussions, Samer Sabri for helpful input with the writing and the anonymous reviewers for helpful comments. This project was partially funded by a UCSD Academic Senate research award to AJY.

## Footnotes

[1]https://www.faceplusplus.com

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
