[Reviews · NeurIPS 2018]

Reviewer 1



Summary of the submission ===================== The authors present their theory of "typicality", to explain why the effect known as Beauty in Averagesness is distorted in some cases, for example, where the viewers previously know the people being judged. The authors introduce a very interesting hypothesis on why complex objects are disliked, in general, when they deviate from individual categorical prototypes of specific objects. The results of the simulations show agreement with the findings in social psychology, suggesting the typicality hypothesis may hold true. Quality ===== The study, hypothesis, methodology and evaluations excel in quality, in my opinion. The work is very interesting and very well framed in the state of the art of attractiveness perception. My critic here is about the brevity of section 6. It seems to be a key point in your theory to be able to prove that disfluency and typicality do not concur, or else, the effect can be as well explained simply as the result of disfluency. The simulation results are interesting. Could you run a perceptual study to validate this claim? The other two cases (familiar faces or race) included blending in the test. It is not clear to me from the text if you are also blending image per age here. Clarity ===== The paper is very well written and easy to follow. And mentioned in my previous point, it would help improve the clarity and quality if the authors would elaborate further on the section 6. Line 123 is confusing. In line 182, the term AAM appears for the first time without explanation. Originality and Significance ====================== The work presented in the paper is original and aims to answer some open questions arising from famous findings in research on attractiveness. Indeed, as the authors motivate early in the paper, perception of attractiveness is crucial in social outcomes and it is highly relevant to investigate why and how it is perceived. The findings of the authors help shed light on the fluidity of attractiveness and its context dependance. Comments after the rebuttal ************************************ As I said before, I like the paper and work a lot. However, as I mentioned before, the key point of your hypothesis is on the differentiation between disfluency and typicality. From your reply, it is not clear to me if this experiment will be actually carried out and included in your results, and just yet it is not convincing enough that this will work. I therefore lower my score.

Reviewer 2



The authors provide a principled statistical account that explains conflicting results in the human perception of faces, and use the AAM technique to translate their approach to a tangible model. They evaluate the account against data from extant experiments. Bridging conflicting existing experimental results with concise and elegant statistical account is definitely a worthy endeavor. Building such a model could lead to a substantial contribution. My impression, however, was that the computational level of the model (section 3 and 4.1), the algorithmic (implementation using AAM) and the interpretation of simulation results in section 4.1.2 are not seamlessly connected. The authors make implicitly strong assumptions to build thee connections, which hinder the overall scope and potential of the advanced approach. Such discrepancies between abstract theories and ways to analyze empirical/simulation data are not uncommon in the social sciences (e.g models in economics tend to be very abstract are often translated to linear equations when tested against empirical data) but are quite uncommon in cognitive science and neuroscience, where typically the computational and the algorithmic level of theories are much better integrated. Minor remarks: 1.The references need to be homogenized. At the moment there are several different styles used. 2. The quality of the graphs could be further improved. Figures 2.b.c They appear to be blurry in my print out. The error bars are quite small. Same for Figure 3. 3. In section 6 the authors propose an experiment and present predictions of two competing accounts. The disfluency account is only mentioned briefly in the introduction before and the details are skipped. I would allocate some supporting material explaining the competing account better, possibly in an appendix. It would have been great to see actual results here, rather than just predictions.

Reviewer 3



Beauty-in-averageness and its contextual modulations: A Bayesian statistical account --Summary-- The paper introduces a model to explain the beauty-in-averageness and ugliness-in-averageness phenomena: The more "typical" a face, the more attractive it is. Typicality of an object is in some sense its information complexity with respect to an underlying model, perhaps conditioned on a specific context. The paper proposes a specific model that validates the proposed explanation for the phenomena, as shown by an empirical study. A face is modeled as a vector of features, which is in turn a mixture of Gaussians: Some Gaussians correspond to features for prototypical faces, whereas other Gaussians correspond to typical features for different categories. This model makes per-category average faces easy to represent, and also faces closer to prototypes. On the other hand, a face that deviates from either prototypes or categories is harder to encode---say an average of two largely distinct category-average faces. The paper proposes an explanation for previously unexplained phenomena that have been subject interest in previous work, and in this sense it seems interesting to me. The claims of the paper are somewhat imprecise and should be clarified, especially so that it is clear what is novel in the paper (see the first comment in the comments section). The main contribution of the paper is a model for faces that explains data & phenomena observed in other studies. The claims that this model is a plausible _predictive_ model for existing data is satisfactorily backed up by the empirical study. The paper is otherwise clear about what it is doing, the results are clear and I am convinced that they can be reproduced. The paper seems a bit vague about potential impact of the result (especially at the discussion in the end), and that should be clarified. --Comments-- Reference 43 seems to quite clearly apply to BiA, and seems more related to the paper than suggested in the literature review. In particular, the idea that compressibility and attractiveness are related is not new and is also present in 43. From the purely abstract view of the model proposed in the submitted paper, there is very little difference between category prototypes and individual prototypes (familiar faces), which somewhat suggests the hypothesis that closeness to prototypes dictates attractiveness (and the beauty-vs-ugliness in average is an observation of presence or absence of average prototype). I found myself confused by the paper's assumptions on the data model---not what they are, but the fact that they are assumptions. It would be more clear to make no assumptions about what is really happening, but rather say "we chose to model the random variable X as ...". In this way, the reader can understand that the conclusions are not vacuous when reality does not conform to the model used. Indeed, the data sort of conforms to the model used and that is enough for the model to be deemed explanatory of different phenomena. In contrast, if the language "we assume that X ~ ..." is used, the reader will either read this as "we model X as" or be confused, asking theirself "what if the assumption is violated?" The paper's contributions to explaining BiA and UiA are interesting, but the paper is a bit vague about other ways the work can have impact (lines 233-246). In particular, the proposed model is effectively a mixture of Gaussians, so it should be clarified how the paper's contributions can be informative to modeling human preferences in other domains, before what mixtures of Gaussians already give us. If space can be spared, I suggest that the paper have some comments about what the current model may fail to capture. For example, we see asymmetry in Fig. 4-e, and we see that uncertainties do not match in Fig. 2-b&c. This will naturally outline directions for future work and will be constructive to the paper's empirical study. The recognition model in Sec. 4.1.1 may be further simplified, by restricting the h_r from the outset to what is described in the "Approximation" paragraph. The point is that if the model used for the empirical study collapses to that, then there is no need to describe "intermediate" model ideas. (One exception is if the intermediate model is more common in the literature, but in this case it would be informative to make a remark about this commonality and about the choice of not using this common model in favor or a simpler one.) For the response, please clarify how your predictive model can have impact in future work. Relevance and potential impact are in this reviewer's opinion an important factor for the 8-10 score band. --Minor comments-- [57] induce a larger [137] |G| = 2 [179] There is a dot below the legend of Figure 1. [193] contains a basis that describes [225] liking'' [Fig 3-d] the distribution is similar to a mixture of normals - Please fix your citation formatting. There are some titles written in all capital letters, and different formats for the same proceedings (including past NIPS proceedings). - The paper seems to make no distinction between non-beauty and ugliness, but that seems to bear a difference in related work, so perhaps it would be useful to remark that Ugliness in Average may be conflated with "Non-Beauty in Average". -- Post Feedback Response -- Thank you for the response. I am happy with the submission and the motivation in the response, and I think they enrich the paper. After the discussion, I am maintaining my original score.